# Analytical Assessment and Nutritional Adequacy of School Lunches in Sintra’s Public Primary Schools

**DOI:** 10.3390/nu13061946

**Published:** 2021-06-05

**Authors:** Telma Nogueira, Raquel J. Ferreira, Vitória Dias da Silva, Mariana Liñan Pinto, Carlos Damas, Joana Sousa

**Affiliations:** 1Laboratório de Nutrição, Faculdade de Medicina, Universidade de Lisboa, 1649-028 Lisboa, Portugal; vitoriasilva@medicina.ulisboa.pt (V.D.d.S.); marianapinto@medicina.ulisboa.pt (M.L.P.); joanamsousa@medicina.ulisboa.pt (J.S.); 2Instituto de Saúde Ambiental, Faculdade de Medicina, Universidade de Lisboa, 1649-028 Lisboa, Portugal; 3Câmara Municipal de Sintra, 2714-501 Sintra, Portugal; raquel.ferreira@cm-sintra.pt; 4Escola Superior de Tecnologia da Saúde de Lisboa, Instituto Politécnico de Lisboa, 1990-096 Lisboa, Portugal; cadamas@estesl.ipl.pt; 5Direção de Qualidade e Ambiente, Indústria e Comércio Alimentar, S.A, 1000-203 Lisboa, Portugal

**Keywords:** school, school lunch, food analysis, child nutrition, food system

## Abstract

School meals present several cost benefits overtime at the short, medium, and long term for individuals and society. This cross-sectional study aims to analyse the nutritional composition and evaluate the adequacy of school lunches. One hundred and fifty-eight samples were collected and analysed from 10 primary schools in Sintra’s municipality, served during one week. On average, energy (27.7% daily energetic requirements) and carbohydrate (48.1%) contents did not reach the reference values, and the content of protein (19.5%) exceeded the reference value (*p* < 0.05). The mean total fat (28.8%) and saturated fatty acids (5.4%) content complied with the recommendations. The mean salt (1.7 g) and dietary fibre (8.3 g) content exceeded the reference value but did not differ significantly from the recommendations. Addressing school canteens is crucial, not only in a nutritional approach, but also as an opportunity to achieve healthier, sustainable, and accessible food systems, aligned with the Sustainable Development Goals 2030. We highlighted the importance of evaluating evidence-based practices and disseminated practice-based evidence regarding the adequacy of school lunches.

## 1. Introduction

During childhood, adequate eating habits are crucial for optimal growth, development, health, and wellbeing, as well as for disease prevention [1,2]. Schools are a privileged environment to protect and support adequate nutrition in children and their families and communities [3,4]. School meals present several cost benefits overtime at the short, medium, and long term for individuals and society [5]. School meals have a great impact on children’s health and wellbeing [6,7]. Providing healthy school meals for all children, regardless of their socioeconomic position, is a recognized strategy for reducing health inequalities [8]. Besides the crucial role of dietary habits in human health, diets are inevitably related to environmental health [9]. Thus, school meals also represent a challenge to environmental sustainability, and they are an opportunity to tackle food waste, protect natural resources and biodiversity, contribute to resilient food systems, and therefore reduce their environmental impact [10,11].

School lunches in Sintra, Portugal, consist of soup (vegetables and/or legumes), a main course (meat, fish, poultry, or a lacto-ovo-vegetarian alternative, served with rice, potatoes, or pasta, and vegetables), dessert (fresh fruit or sweetened dessert once a week), bread, and water as a beverage.

Children are next-generation consumers, so influencing their eating behaviours could have an impact on the sustainability of the food systems [12,13], as the acquisition of more sustainable habits could continue into adulthood, reducing the amount of waste produced by future generations [12]. Additionally, a healthy school food environment, where the food provided is accessible, enjoyable, safe, and nutritious, enables and encourages the entire school community to make healthy food choices [11].

Dealing with nutrition concerns in childhood requires coherent action in the school setting, so the development of food and nutrition policies are central [3,14]. Several national [15,16,17] and international [3,18,19] guidelines that aim to regulate and improve the school food environment have been developed. In the past two decades, entities, such as the World Health Organization [20,21], European Commission [22], and Food and Agriculture Organization of the United Nations [23,24,25], have established action plans to respond to emerging human and environmental health challenges. The signing of the Milan Urban Food Policy Pact reinforces the responsibility of local governments due to their proximity to the community in promoting resilient and sustainable food systems capable of providing adequate nutrition [24]. Governments can take various actions to create a healthy and sustainable school environment, namely setting nutritional standards for school meals, minimizing the exposure to advertisements of foods and beverages high in fat, sugars, and salt [26], counteracting food waste, and setting up school gardens [11]. 

It has been recognized that the third United Nations Sustainable Development Goal (good health and wellbeing) involves a broad range of social determinants covered by the remaining goals [27]. Sustainable and healthy food systems are directly or indirectly connected to all of the Sustainable Development Goals when all parts of society are seen as interconnected sectors of the biosphere [28]. Therefore, ensuring an adequate school food environment contributes to an integrated response to the United Nations 2030 Agenda for Sustainable Development [24,29].

School meals are crucial to promote healthy and sustainable food behaviours amongst the school community. Nevertheless, data of Portuguese school lunches content and their adequacy are scarce. Thus, this study aims to analyse the nutritional composition and evaluate the adequacy of school lunches served at Sintra’s public primary schools’ canteens.

## 2. Materials and Methods

### 2.1. Study Design, Population, and Sampling 

This analytical observational study was conducted in the municipality of Sintra, Portugal. Sintra’s municipality oversees the management of public school meal service in 90 establishments (67 with local confection and 23 with a deferred cook/chill catering system) [30], providing approximately 15,000 school lunches a day. Thus, there are two established lunch menus: a local confection menu and a deferred cook/chill menu (Appendix A and Appendix B). The school lunch menus are conceived by registered dietitian/nutritionists considering the nutritional needs of the school population, with regards to the guidelines of the Directorate-General for Education [31]. These guidelines establish the nutritional quality and quantity of the school menu, including the meal composition, portion size, cooking methods, and food sustainable principles [31]. Sintra’s municipality has established a cycle of a 15-week school lunch menu for schools with local confection and a cycle of an eight-week school lunch menu for the deferred cook/chill menu. The selection of school canteens for sampling was done by convenience and consisted of school lunch menus from nine primary schools with local confection and from one primary school with a deferred cook/chill catering system. This analysis corresponds to one week of school lunch menus in each school.

### 2.2. Data Collection

In this study, 158 samples were collected and analysed from 10 primary schools from Sintra’s municipality, served for one week between 4–8 June 2018. From the 50 school lunches, one soup and one main course was lost during the data collection. In each school, samples of soup, the main course, dessert, and bread in the portions that were served to the children were taken. In each school, samples were daily collected and frozen by the canteen manager. Afterwards, laboratory technicians transported the samples to an accredited and independent laboratory. Disposable gloves and masks, precision balance, serving bowls, spoons, disposable paper, and plastic bags were used to collect the samples. It was guaranteed that the sampling area was a properly hygienic environment. The nutritional composition of the samples was obtained by analytic methods conducted by an independent and accredited laboratory considering the techniques presented in Table 1. The school lunches were analysed for net weight (g), energy (kcal), protein (g), carbohydrates (g), fats (g), saturated fatty acids (SFA) (g), dietary fibre (g), sodium (g), and salt (g) content. The energy value was calculated using the conversion factors established in Regulation (EU) No 1169/2011. The salt content was calculated considering the formula salt = sodium × 2.5. The carbohydrate content was obtained through calculation by difference [32].

### 2.3. Assessment of Nutritional Adequacy

It was considered that lunch should contribute between 30 to 35% of the daily energy requirements (DER) [33]. The DER of six- to 10-year-old children with different physical activity profiles is 1640 kcal per day, according to the American guidelines [34]. Thus, lunch should contribute between 492 kcal (30% DER) to 574 kcal (35% DER) [34]. To calculate protein, carbohydrates, lipids, and SFA requirements, it was considered that lunch should contribute 30% of the recommended DER, following the methodology used in the national guidelines for school menus [35]. According to the methodology referred to [35], the percentage distribution for one dietary day was considered, according to the World Health Organization (WHO) recommendations [36]: protein 10 to 15% (12 g to 18 g), carbohydrates 55 to 75% (68 g to 92 g), total lipids 15 to 30% (8 g to 16 g), and SFA < 10% (<5.5 g). To calculate dietary fibre adequacy, it was considered 30% of the total daily dietary fibre value recommended [37]. Thus, the minimum content of dietary fibre was 6.89 g at lunch, considering the daily recommendations of 14 g per 1000 kcal [33,34]. The WHO recommends reducing sodium intake in children to <2 g/day of sodium, which corresponds to <5 g/day of salt [38]. Therefore, the amount of salt in the school lunches was calculated considering 30% of the recommended daily salt value, corresponding to <1.5 g.

### 2.4. Statistical Analyses

Statistical analysis was performed using the Statistical Package for Social Sciences version 24 (IBM Corp, Armonk, NY, USA) and the RStudio version 3.6.3 (RStudio: Integrated Development for R. RStudio, PBC, Boston, MA, USA). The normality distribution of data was verified by the Shapiro–Wilk test. Descriptive statistics were used to characterise the sample expressed by absolute and relative frequencies, for qualitative variables, and mean or median and standard deviations for quantitative variables. Two-sided, one-sample *t*-tests were used to test whether the energetic (30% of DER) and nutritional content means differed from the reference values (the lowest value in the reference interval was chosen, except for total fat, where the highest reference value was used). For the other nutrients with a set reference value (dietary fibre, SFA, and salt), the hypotheses tested were that the mean content did not exceed the reference value. The results were considered significant for a 5% significance level (*p* < 0.05).

## 3. Results

For one week, 50 school lunches from 10 schools (158 samples) equivalent to the portion made available to children were collected and analysed by analytical methods. The final sample was composed of 49 soups, 49 main courses (19 meat dishes, 20 fish dishes, and 10 lacto-ovo-vegetarian dishes), 50 desserts, and 10 breads (one for each school). The weight and nutritional composition of the school lunch samples are presented in Table 2. 

The mean content of energy and nutrients compared to the reference values are presented in Table 3. The mean content of energy and carbohydrates did not reach the reference values, and the content of protein exceeded the reference value (*p* < 0.05). Mean salt and dietary fibre content exceeded the reference value, but did not differ significantly from the recommendations. Mean total fat and SFA content complied with the recommendations. 

## 4. Discussion

In agreement with previous studies [39,40,41], the content from the school lunches did not reach the reference values included in the guidelines for carbohydrates, but exceeded the reference values for protein. School lunches provided on average of 27.7% of the DER, which is slightly less than the 30% stated in the guidelines. The insufficient energy content is mainly due to the insufficient amount of carbohydrates provided. 

The mean content of salt did not differ significantly from the reference values, unlike other national studies, where the mean content of salt exceeded recommendations [41,42]. This result might be due to the salt control policy already implemented in school canteens in Sintra [43]. Concerning salt content, the bread alongside the meal may contribute to an increase in the salt content of school lunches. The analytical analysis shows that the mean salt content in bread (0.3 ± 0.1 g in 20 g of bread) is higher than the limit defined by the Portuguese legislation (1.4 g of salt per 100 g of bread) since 2009 [44]. Additionally, a collaboration protocol between the Directorate-General for Health, the National Institute of Health Doutor Ricardo Jorge, and the Associations of Bakery, Pastry and Similar Industries that goes beyond the current legislation established new goals for the reduction of salt in bread, with the ultimate goal of a maximum content of 1.0 g of salt per 100 g of bread. As intermediate targets, the following values were established: 1.3 g of salt per 100 g in 2018, 1.2 g of salt per 100 g of bread in 2019, and 1.1 g of salt per 100 g of bread in 2020 [45]. It is well-established in the literature the relationship between the excessive consumption of salt and non-communicable diseases [46]. Thus, the reduction of salt in bread is one of the key aspects to decrease daily salt consumption, since bread is generally manufactured by bakeries in which consumers have low power to choose [47]. Therefore, this result reinforces the importance of monitoring and evaluating food and nutritional policies. In addition, we must consider that the presence of tofu—a product with added salt—contributed to an increase the mean salt content, even though the mean salt content did not differ significantly from the recommendations. Tofu is only occasionally present on the school menu in Sintra—once in every 15 weeks—as recommended [48].

Although the content of dietary fibre shows disparities between lunches, the mean content did not differ significantly from the reference values. 

At the present study, the mean total fat and SFA complied with the recommendations. Regarding mean total fat, this result is concordant with another study [39], while the SFA result is not. 

Actually, Portugal has been facing a diet transition, moving away from the traditional Mediterranean diet to a more Westernized diet [49]. Our results are aligned with national data about food availabilities [50], where the availability of the food group of meat, fish, and eggs was above the recommended consumption, and fruits and horticultural were below the recommended consumption. Furthermore, Portuguese National Food and Physical Activity Survey data [51] pointed out certain nutrients and foods in need of improvement. It was found that 83.2% of children have a daily intake of protein greater than 2 g/kg weight. In children under 10 years old, it was described that the most consumed food group is meat, fish, and eggs. In addition, 16.3% of children ingest more than 100 g/day of red meat [37]. By contrast, cereals, fruits, and vegetables are the least consumed [51], with 72.0% of children not reaching WHO recommendations for fruit and vegetable consumption [36,51].

School meals have a great impact on children’s health and wellbeing [6,7]. When compared with lunches prepared and/or consumed out of school, school lunches present better nutritional composition [52,53]. 

Although the average energy content of school lunches is below the recommendations, empirical knowledge by local canteen staff indicates the occurrence of plate waste. Thus, we cannot exclude the possibility that the canteen staff may serve portions that differ from recommendations, according to children’s food preferences, to reduce plate waste. Moreover, the literature indicates that primary school children’s snacks are mainly composed of foods and beverages high in fat, sugars, and salt, favouring a high energy contribution to children’s daily energy needs [54,55]. This excessive dietary consumption before lunch could decrease children’s hunger sensation for school lunches and contribute to plate waste.

Despite guidelines suggesting a contribution of 30% of DER at lunch [33], there is substantial plate waste in school canteens [12,56,57,58] and a high percentage of overweight school-aged children [51,59]. Thus, we reflected whether guidelines regarding the ideal contribution of each meal in DER should be reviewed for children.

Hence, it is well understood the implications of school lunches in the nutritional and overall health of the school community, as well as their impact in food systems. Thus, a perfect nutritionally designed menu is unworthy if there is substantial plate waste [60]. This becomes particularly critical when considering that different food types and healthier ingredients, specifically fruit and vegetables, are often the most wasted [13,60,61,62,63], influencing nutritional adequacy, mainly of micronutrients and dietary fibre. We suggest including children’s lunch as pedagogical time in teachers’ labour schedule. The supervision of teachers in school canteens is crucial, being capable of reinforce this setting as central in nutrition and health education, applying in practice knowledge, skills, and behaviours about healthy eating.

The authors are aware of the study limitations. The menu consists of 15 rotating weeks, and this analysis corresponds to one week out of the 15. In the future, it would be relevant to include a larger sample size and analytical analyses of other micronutrients, namely potassium, iron, iodine, zinc, calcium, and vitamin D to fully characterize the nutritional adequacy of school lunches [39,64]. 

A major strength of this analysis is related to the accuracy of the materials and methods used to evaluate the nutritional composition of school lunches. Samples were analysed in an accredited laboratory; consequently, the results are not an assumption of the quantities present in the lunch technical sheets, nor the nutritional value based on nutritional composition tables present in books or reports. Moreover, the samples analysed corresponded to the actual portion served to children. Thus, the procedures defined for sampling, as well as the reliable methods for analysis of the school lunches, ensures more accuracy and a good reproducibility of the study. Furthermore, the assessment of school lunch nutritional adequacy was established considering international recommendations for the age group under study [33,34]. 

The implementation of joint school food environment policies and food and nutrition education interventions are more efficient in enhancing children’s dietary habits [11]. Sintra’s municipality has recognized the role of such complementary policies and implements, and monitors a comprehensive and multicomponent school-based intervention: Sintra Grows Healthy [43]. Thus, ensuring coherent action between the integration of food literacy in the curriculum and the development of actions capable of modifying the school food environment is crucial to facilitate and support the adoption of healthy behaviours. 

## 5. Conclusions

This study analysed the nutritional composition of school lunches in the primary schools of Sintra’s municipality and indicated the need of improvement in the content of energy, carbohydrates, and protein when compared to current recommendations. Addressing school canteens is crucial, not only in a nutritional approach, but also as an opportunity to achieve healthier, sustainable, and accessible food systems aligned with the Sustainable Development Goals 2030. We highlight the importance of evaluating evidence-based practices and disseminating practice-based evidence regarding the adequacy of school lunches.

## Figures and Tables

**Table 1 nutrients-13-01946-t001:** Analytical methods.

Parameter	Method
Net weight (g)	Gravimetry
Energy (kcal)	Calculation based in Regulation (EU) No 1169/2011
Protein (g)	Dumas technique
Carbohydrates (g)	Calculation based in Regulation (EU) No 1169/2011
Fats (g)	Pulsed nuclear magnetic resonance
SFA (g)	Gas Chromatography with flame-ionization detection
Dietary Fibre (g)	Enzymatic/gravimetric
Sodium (g)	Flame atomic absorption spectroscopy
Salt (g)	Calculation from sodium content based in Regulation (EU) No 1169/2011

SFA: Saturated fatty acids.

**Table 2 nutrients-13-01946-t002:** Weight and nutritional composition of main course, soup, dessert, and bread of school lunches analysed.

	Per Dose	Mean	SD	MIN	MAX
MAIN COURSE (*n* = 49)	Weight (g)	233.9	56.4	132.4	359.0
	Energy (kcal)	277.7	285.9	131.3	504.0
	Carbohydrates (g)	24.9	7.5	12.2	43.5
	Protein (g)	17.3	7.0	5.3	40.9
	Total fat (g)	11.2	6.7	2.4	31.9
	SFA (g)	1.9	0.2	0.7	4.4
	Dietary Fibre (g)	4.2	3.6	0.8	18.0
	Salt (g)	1.0	0.4	0.4	2.2
SOUP (*n* = 49)	Weight (g)	211.0	35.9	137.9	280.3
	Energy (kcal)	66.0	12.5	41.4	95.1
	Carbohydrates (g)	8.9	2.1	4.8	14.3
	Protein (g)	1.8	0.7	1.0	4.5
	Total fat (g)	2.2	0.5	1.3	3.3
	SFA (g)	0.5	0.1	0.3	0.7
	Dietary Fibre (g)	1.9	0.7	1.0	4.8
	Salt (g)	0.4	0.6	0.2	0.9
DESSERT (*n* = 50)	Weight (g)	93.4	14.9	52.4	132.9
	Energy (kcal)	57.3	20.8	24.7	109.0
	Carbohydrates (g)	10.8	4.5	4.0	21.1
	Protein (g)	1.01	0.7	0.3	3.0
	Total fat (g)	0.8	0.2	0.4	1.4
	SFA (g)	0.2	0.2	0.0	0.9
	Dietary Fibre (g)	1.5	0.8	0.3	3.6
	Salt (g)	<0.1	<0.1	<0.1	0.1
BREAD (*n* = 10)	Weight (g)	20.0	0.0	20.0	20.0
	Energy (kcal)	54.3	0.7	53.2	56.8
	Carbohydrates (g)	10.4	0.2	10.0	10.8
	Protein (g)	2.0	0.2	1.7	2.3
	Total fat (g)	0.4	<0.1	0.4	0.4
	SFA (g)	0.1	<0.1	0.1	0.1
	Dietary Fibre (g)	0.6	0.1	0.5	0.8
	Salt (g)	0.3	0.1	0.3	0.4

SD: standard deviation; MIN: minimum; MAX: maximum; SFA: saturated fatty acids.

**Table 3 nutrients-13-01946-t003:** Weight and nutritional composition of total meal of school lunches analysed and compared to the reference value.

Per Dose			Reference Values	Mean	SD	MIN	MAX	95% CI (Lower; Upper)	*p* ^c^
TOTAL MEAL ^a^ (*n* = 48)	Weight	g	NA	556.7	73.9	429.1	710.4	NA	NA
	Energy	kcal	492–574	454.4	95.3	282.1	690.0	399.4; 473.1	0.004 *
		% DER ^b^	30–35	27.7	5.8	17.2	42.1	24.4; 28.9	
	Carbohydrates	g	68–92	54.7	9.0	36.6	71.3	48.6; 56.5	<0.001 *
		% E	55–75	48.1	7.3	29.7	58.0	39.5; 45.9	
	Protein	g	12–18	22.1	7.4	9.1	47.4	18.8; 23.6	<0.001 *
		% E	10–15	19.5	6.0	7.4	38.5	15.3; 19.2	
	Total fat	g	8–16	14.5	7.0	5.0	35.4	11.9; 16.1	0.058
		% E	15–30	28.8	12.8	9.1	64.8	21.7; 29.4	
	Saturated Fatty Acids	g	5.5	2.7	0.3	1.2	5.2	2.5; 3.3	<0.001
		% E	<10	5.4	0.6	2.2	9.6	6.6; 6.0	
	Dietary Fibre	g	6.9	8.3	3.9	4.5	22.6	6.8; 9.1	0.078
	Salt	g	<1.5	1.7	0.4	1.0	3.0	1.5; 1.8	0.084

SD: standard deviation; MIN: minimum; MAX: maximum; CI: confidence interval; DER: daily energy requirements; % E: Energy standardized considering 30% DER; NA: Not applicable. ^a^ Total meal: soup + main course + bread + dessert; ^b^ energy of total meal is presented in % DER of six- to 10-year-old children (1640 kcal per day, according to the American guidelines). ^c^ Since two-sided, one-sample *t*-tests were used, deviations of the estimated parameter are detected in either direction. Therefore, only significant differences that deviate from the recommendations are marked with an asterisk. * Indicates significant differences that deviate from the recommendations.

## Data Availability

The data presented in this study are available on request from the corresponding author.

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
