# Peer review of "Analytical Assessment and Nutritional Adequacy of School Lunches in Sintra’s Public Primary Schools"

_nutrients, 2021, doi:10.3390/nu13061946_

Round 1
Reviewer 1 Report
- Overall description of the manuscript
This cross-sectional study collected 158 school lunch samples of 1 week from 10 primary schools in Sintra’s municipality. The contents of energy, major nutrients, saturated fatty acids, dietary fiber and sodium were analyzed for both lunch and items (main course, soup, desserts and bread) in the menu. The analyzed results were compared with reference values to evaluate the adequacy of each food components. The reference values include US IOM for energy, national guidelines for school lunch, and WHO recommendations.
- Comments
- This manuscript provides very limited data to evaluate the nutritional adequacy of the lunch collected. Since children need not only adequate energy and proper distribution of macronutrients but also enough of micronutrients for growth and development, the data of this study providing no information regarding micronutrients may not be able to help readers gaining insight into the issues of child nutrition/education and their lunch contents.
- Authors mentioned the importance of SDG 3 and the crucial role of school meal service in implementing SDG 3 (line 66-77). However, the data reported in the manuscript could not reflect or support any actions related to SDG 3.
- The 7th and 9th paragraphs (line 205-213, 218-227) of “Discussion” discusses the plate waste, sugary beverage and improper snack consumption and their impacts on food intake of children. However, there are no data reported to these issues. And these issues seem not relevant to the title of this manuscript.
- The 8th paragraph (line 214-217) of “Discussion” suggests the possible need of revising the national guidelines for school lunch because of substantial plate waste and high percentage of overweight children. Guidelines are supposed to have their scientific bases; the problems of plate waste and child obesity need to be solved through other strategies. The limited amount of data may not support authors to raise such a suggestion.
- Suggestions to authors:
(1) In order to reveal in-depth details regarding the nutritional adequacy of collected school lunch, authors may consider calculate the nutrient contents (especially vitamins and minerals) using nutrient data base if these analyzed data are not available.
(2) Nutritional adequacy of meals may also be assessed through food-based approaches. Since school lunch menu designed need to cohere the national guidelines, and authors mentioned there exists a dietary pattern transition moving away from traditional Mediterranean diet to Westernized diet, authors may consider using some dietary assessment tools reported in the literatures, such as DDS (dietary diversity score), HEI (healthy eating index), MDS (Mediterranean diet score), etc. to show the quality of school lunch designed for the children of public primary schools.
(3) With the further analyses suggested in (2), authors may gather more information to discuss if SDG 3 could be implemented in the schools receiving lunch services.
Author Response
Thank you for the opportunity to revise our paper on ‘Analytical assessment and nutritional adequacy of school lunches in Sintra’s public primary schools’. The suggestions offered have been greatly helpful.
Comment: This manuscript provides very limited data to evaluate the nutritional adequacy of the lunch collected. Since children need not only adequate energy and proper distribution of macronutrients but also enough of micronutrients for growth and development, the data of this study providing no information regarding micronutrients may not be able to help readers gaining insight into the issues of child nutrition/education and their lunch contents.
Response: Thank you for pointing this out. As stated in lines 235-238 we are aware of this issue. The analytical methods performed in an accredited laboratory are very expensive, so our priority was to evaluate Energy, Protein, Carbohydrates, Fats, SFA, Dietary Fibre, Sodium, and Salt. Also, it was only evaluated one meal (lunch) of the entire day and there is currently (at least in Portugal) any guideline suggesting the adequate distribution of micronutrients specifically at lunch.
Comment: Authors mentioned the importance of SDG 3 and the crucial role of school meal service in implementing SDG 3 (line 66-77). However, the data reported in the manuscript could not reflect or support any actions related to SDG 3.
Response: Thank you for your comment. There is evidence that school meal service is an essential element to contribute to the SG3 – “Ensuring healthy lives and promoting well-being at all ages”. School meals have a great impact on children’s health and well-being. A healthy school food environment enables and encourages the entire school community to make healthy food choices. Therefore, ensuring an adequate school food environment contributes for an integrated response to the United Nations 2030 Agenda for Sustainable Development. Literature also indicates that the implementation of jointly school food environment policies and food and nutrition education interventions are more efficient in enhancing children’s dietary habits. Sintra’s municipality recognizes this and implements and monitors a comprehensive and multicomponent school-based intervention – Sintra Grows Health – ensuring a coherent action between the integration of food literacy in the curriculum, and the development of actions capable of modifying the school food environment, crucial to facilitate and support the adoption of healthy behaviours and achieving the sustainable development goals.
Comment: The 7th and 9th paragraphs (line 205-213, 218-227) of “Discussion” discusses the plate waste, sugary beverage and improper snack consumption and their impacts on food intake of children. However, there are no data reported to these issues. And these issues seem not relevant to the title of this manuscript.
Response: Thank you for pointing this out. The literature indicates that primary school children’s snacks are mainly composed of foods and beverages high in fat, sugars, and salt, favoring a high energy contribution to children’s daily energy needs (references 54,55). Also, we are already working on data, that is not yet published, related to children’s nutritional snacks composition in Sintra, that show an average intake of the triple of the recommended. Regarding plate waste at school canteens, there are several studies pointing this emergent problem (references 12, 56 -58).
Comment: The 8th paragraph (line 214-217) of “Discussion” suggests the possible need of revising the national guidelines for school lunch because of substantial plate waste and high percentage of overweight children. Guidelines are supposed to have their scientific bases; the problems of plate waste and child obesity need to be solved through other strategies. The limited amount of data may not support authors to raise such a suggestion.
Response: Thank you for the observation. We absolutely agree that plate waste and child obesity must be solved through other strategies and we did not intend to suggest that revising the guidelines was a way of solving these issues. However, we were raising possible hypotheses that could address it. Nevertheless, to avoid misinterpretation we have clarified that sentence (line 222-223).
Suggestions to authors: In order to reveal in-depth details regarding the nutritional adequacy of collected school lunch, authors may consider calculate the nutrient contents (especially vitamins and minerals) using nutrient data base if these analyzed data are not available.
Response: Thank you for your comment. Our analytical analyses did not included micronutrients, so we cannot add them. Estimating using nutrient data bases would not reflect an accurate micronutrient content of the school lunches and would compromise the major strength of this analysis related to the accuracy of materials and methods used to evaluate the nutritional composition of school lunches.
Suggestions to authors: Nutritional adequacy of meals may also be assessed through food-based approaches. Since school lunch menu designed need to cohere the national guidelines, and authors mentioned there exists a dietary pattern transition moving away from traditional Mediterranean diet to Westernized diet, authors may consider using some dietary assessment tools reported in the literatures, such as DDS (dietary diversity score), HEI (healthy eating index), MDS (Mediterranean diet score), etc. to show the quality of school lunch designed for the children of public primary schools.
Response: Thank you for the suggestion. We believe it would be of the utmost relevance to assess meals from that approach but in another manuscript as our goal with this work was to assess meals based on the analytical analyses performed. Also, in Sintra the school lunches’ menu are conceived by registered dietitian/nutritionists considering the nutritional needs of the school population, following the guidelines of the Directorate General for education and based on the Mediterranean diet. However, as stated in our work, a perfect nutritionally balance meal is unworthy if children do not consume it, that was why we evaluated the actual portion served to children instead of the recommended portion.
Suggestions to authors: With the further analyses suggested in (2), authors may gather more information to discuss if SDG 3 could be implemented in the schools receiving lunch services.
Response: Thank you for the suggestion.
Reviewer 2 Report
The authors have done a quite elegant study on the composition of school food lunch in Sintra (Portugal). The research has been clearly designed and performed with quite clear aims.
Two minor comments:
- Provide some data on the content of the guidelines of the Diretctorate General for education regarding school menus.
- Lines 105-108: some values were not obtained by analysis, but estimated. Provide a minimal explanation on how these values were obtained (according to the specific content of the dish for example).
My congratulations
Author Response
Thank you for the opportunity to revise our paper on ‘Analytical assessment and nutritional adequacy of school lunches in Sintra’s public primary schools’. The suggestions offered have been greatly helpful.
Comment: Provide some data on the content of the guidelines of the Diretctorate General for education regarding school menus.
Response: Thank you for helping us improving our manuscript. We added some information referring to the guidelines content (lines 87-89).
Comment: Lines 105-108: some values were not obtained by analysis, but estimated. Provide a minimal explanation on how these values were obtained (according to the specific content of the dish for example).
Response: Thank you for pointing this out. We provided some explanation on how these values were obtained (lines 110-113).